# Potential Application of Plant-Derived Compounds in Multiple Sclerosis Management

**DOI:** 10.3390/nu16172996

**Published:** 2024-09-05

**Authors:** Seth Woodfin, Sierra Hall, Alexis Ramerth, Brooke Chapple, Dane Fausnacht, William Moore, Hana Alkhalidy, Dongmin Liu

**Affiliations:** 1Department of Biology and Chemistry, School of Health Sciences, Liberty University, Lynchburg, VA 24515, USA; sfwoodfin@liberty.edu (S.W.); shall199@liberty.edu (S.H.); aramerth@liberty.edu (A.R.); bechapple@liberty.edu (B.C.); 2Department of Biology, School of Sciences and Agriculture, Ferrum College, Ferrum, VA 24088, USA; dfausnacht@ferrum.edu; 3Department of Human Nutrition, Foods and Exercise, College of Agriculture and Life Sciences, Virginia Tech, Blacksburg, VA 24061, USA; hkhaldi@vt.edu; 4Department of Nutrition and Food Technology, Faculty of Agriculture, Jordan University of Science and Technology, P.O. Box 3030, Irbid 22110, Jordan

**Keywords:** multiple sclerosis, inflammation, plant-derived compounds, polyphenols, alkaloids, terpenoids, catechol

## Abstract

Multiple sclerosis (MS) is a chronic autoimmune disorder characterized by inflammation, demyelination, and neurodegeneration, resulting in significant disability and reduced quality of life. Current therapeutic strategies primarily target immune dysregulation, but limitations in efficacy and tolerability highlight the need for alternative treatments. Plant-derived compounds, including alkaloids, phenylpropanoids, and terpenoids, have demonstrated anti-inflammatory effects in both preclinical and clinical studies. By modulating immune responses and promoting neuroregeneration, these compounds offer potential as novel adjunctive therapies for MS. This review provides insights into the molecular and cellular basis of MS pathogenesis, emphasizing the role of inflammation in disease progression. It critically evaluates emerging evidence supporting the use of plant-derived compounds to attenuate inflammation and MS symptomology. In addition, we provide a comprehensive source of information detailing the known mechanisms of action and assessing the clinical potential of plant-derived compounds in the context of MS pathogenesis, with a focus on their anti-inflammatory and neuroprotective properties.

## 1. Introduction

Multiple sclerosis (MS) is a chronic autoimmune disorder of the central nervous system (CNS) characterized by inflammation, demyelination, and neurodegeneration [1]. Its increasing prevalence positions it as the most common progressive neurologic pathology among young adults globally [2]. Multiple sclerosis typically manifests through a variety of neurological symptoms, including motor dysfunction, sensory deficits, fatigue, and cognitive impairment, leading to significant disability and reduced quality of life [3].

The etiology of MS involves complex interactions between genetic predisposition, environmental factors, and dysregulated immune responses [4]. Although the exact cause remains elusive, it is widely accepted that aberrant immune activation, particularly by autoreactive T cells, plays a pivotal role in initiating and perpetuating the inflammatory cascade within the CNS [5]. This immune dysregulation leads to the infiltration of immune cells across the blood-brain barrier (BBB), resulting in the release of pro-inflammatory cytokines, oxidative stress, and subsequent damage to myelin sheaths and neurons [6].

Current therapeutic strategies for MS primarily focus on modulating immune responses to reduce inflammation and halt disease progression [7]. Current treatments, including disease-modifying therapies (DMTs), symptomatic treatments, acute relapse treatments, rehabilitation therapies, and lifestyle modifications, have shown efficacy in mitigating relapses and delaying the onset of disability [8]. However, their long-term safety profiles and limited effectiveness in progressive forms of MS underscore the need for alternative treatment modalities [8].

Recently, especially in the past decade, there has been growing interest in exploring the therapeutic potential of plant-derived compounds for managing MS-associated inflammation [9,10]. Phytochemicals, including alkaloids, phenylpropanoids, and terpenoids, have demonstrated anti-inflammatory, antioxidant, and potential neuroprotective properties in both pre-clinical and clinical studies [11]. These compounds exert their effects through various mechanisms, including inhibiting pro-inflammatory mediators, modulating immune cell function, and promoting neuroregeneration.

A recently published review focused on the molecular mechanism of some polyphenols’ protective benefits against MS with a focus on the role of gut microbiota in MS etiopathogenesis [12]. This review aims to provide a comprehensive overview of the pathophysiology of MS, with a particular emphasis on the role of inflammation in disease progression. Furthermore, we will critically evaluate the emerging evidence supporting the use of select plant-derived compounds as adjunctive therapies to attenuate inflammation-induced MS symptomology. By elucidating the underlying mechanisms of action and assessing the clinical efficacy of these natural agents, it will encourage the development of novel therapeutic strategies that offer greater efficacy and tolerability for individuals living with MS.

## 2. Molecular and Cellular Basis of MS Pathogenesis

### 2.1. Immune Cell Dynamics in the Brain and Central Nervous System (CNS)

Antigen presentation is central to immune surveillance in the brain. In this process, antigen presenting cells (APCs) sample and present CNS-derived antigens to helper T cells, modulating immune responses [13]. Regulatory mechanisms within the CNS, including regulatory T cells, soluble immunomodulatory factors, and the BBB, tightly regulate immune activation and tolerance, ensuring immune homeostasis within the CNS microenvironment [14]. In this section, we will provide an overview of these normal immune responses, as they are crucial for understanding the pathogenesis of MS.

### 2.2. Antigen Presenting Cells and the Antigen

Upon encountering foreign or self-derived antigens, APCs, such as dendritic cells, macrophages, or B cells, utilize pattern recognition receptors (PRRs) to identify and internalize the antigen through a process known as phagocytosis [15]. Following antigen internalization, the APC processes the antigens into smaller peptide fragments within a specialized intracellular compartment called a phagolysosome, which is the product of the fusion of a phagosome and a lysosome [16]. These peptide fragments are then loaded onto major histocompatibility complexes (MHC), forming MHC–peptide complexes [17]. Through antigen presentation, APCs effectively display these MHC–peptide complexes on their cell surface, where they can be recognized and engaged by T cell receptors (TCRs) expressed on the surface of helper T cells [17]. The interaction between MHC–peptide complexes and TCRs initiates a cascade of immune responses, including T cell activation, proliferation, and differentiation, which collectively tailor the adaptive immune response to combat the specific antigen that was initially encountered by the APC [18].

### 2.3. Neuroinflammation and Demyelination: Molecular Mimicry

In-depth investigations, tracing back to seminal studies from 1935, have sought to explore the similarities and differences between MS and its experimental autoimmune encephalomyelitis (EAE) model [19]. Initially, certain types of immune cells were perceived as the main instigators of CNS demyelination, with specific subsets of CD4+ T cells having been believed to multiply in the brain and release cytokines that harm myelin [20]. However, recent findings have challenged this viewpoint, suggesting that additional immune cell populations, including macrophages, CD8+ T cells, and B cells, also play significant roles in the inflammatory response seen in both EAE and MS lesions [5,21].

Contrary to previous assumptions, doubts have been raised about the harmful effects of certain cytokines on myelin in MS lesions, with emerging evidence pointing towards the potential importance of cytotoxic CD8+ T cells in demyelination [22]. These cells can identify oligodendrocytes and/or myelin antigens due to their expression of specific molecules under inflammatory circumstances, implying a primary involvement in an antigen-focused inflammatory reaction [23]. Moreover, B lymphocytes have emerged as notable contributors to MS pathology, as evidenced by therapeutic trials targeting molecules on these cells, resulting in decreased lesion formation and relapses [24].

Overall, this developing comprehension of MS pathology emphasizes the roles of CD8+ T cells and B lymphocytes in addition to that of CD4+ T-helper cells [25]. Additionally, interactions between T cells and certain glial cells, particularly oligodendrocytes, are crucial for CNS balance and autoimmune diseases like MS [26]. The concept of molecular mimicry adds complexity to this interaction, as foreign substances that resemble CNS antigens can incite self-directed T cell responses, perpetuating neuroinflammatory processes [27].

### 2.4. Inflammatory Cytokine Pathways Involved in MS Pathogenesis

It is thought that the dysregulation of pro-inflammatory cytokine production, including IL-1, IL-6, IFN-γ, and TNF-α, contributes to the pathogenesis and progression of MS [28,29]. These cytokines are often produced in excess and contribute to chronic inflammation, demyelination, and neurodegeneration within the CNS [30,31]. These cytokines collectively promote the recruitment and activation of both helper T cells and macrophages in the CNS [32]. This leads to the formation of inflammatory lesions, breakdown of the BBB, and subsequent damage to myelin and neurons [33]. IFN-γ, primarily produced by activated T cells, further exacerbates inflammation and tissue damage by activating microglia, leading to the release of cytotoxic factors and oxidative stress [34]. Additionally, dysregulated cytokine signaling disrupts the balance between pro-inflammatory and anti-inflammatory mediators, contributing to the chronic and relapsing-remitting nature of MS [35,36,37]. Targeting these dysregulated cytokines through immunomodulatory therapies represents a promising approach for the treatment of MS as this would attenuate inflammation, protect neuronal integrity, and ultimately improve clinical outcomes for affected individuals [38].

#### 2.4.1. TNF-α Signaling

TNF-α is produced by macrophages, monocytes, and T lymphocytes in response to various stimuli such as infection, inflammation, or tissue injury [39]. The production of TNF-α is tightly regulated at the transcriptional and post-transcriptional levels and is only upregulated when the aforementioned conditions are met [40]. The effect of TNF-α as a binding substrate depends largely on the tissue type and TNF-α receptor to which the cytokine binds [39]. To date, there are two known types of TNF-α receptors, TNFR1 (also known as p55 or CD120) and TNFR2 (also known as p75 or CD120b) [41]. TNFR1 is widely expressed and primarily involved in mediating pro-inflammatory and apoptotic signals, while TNFR2 is expressed on immune cells and contributes to immune regulation and tissue repair processes [42]. Under the same convention, the binding of TNF-α to either of these receptors will elicit unique signaling mechanisms, coined as the canonical and non-canonical pathways [43].

The canonical pathway, primarily associated with TNFR1 signaling, begins with the receptor trimerization and assembly of a multiprotein signaling complex along the intracellular domain of TNFR1 [44]. Within this complex, there is an array of adaptor proteins, the first of which is TNF-α receptor-associated death domain (TRADD), as both this adaptor protein and the receptor itself share complimentary death domains [45] (Figure 1). In a similar fashion, receptor-interacting protein-1 (RIP-1) possesses a death domain that allows its articulation with the upstream receptor complex [46]. Additionally, RIP-1 can undergo both polyubiquitination (K63-linked chains) and phosphorylation (S14, S15, S20, and S166) through autophosphorylation and feedback mechanisms from downstream/crosstalk interactions (IκB kinase β and associated TRAFs) [47,48]. Regardless of the catalyst, phosphorylation along these serine residues leads to the activation of RIP-1 as a serine/threonine kinase [48]. TNF-α receptor-associated factor 2 (TRAF2) is then recruited to the growing signaling complex and activated via RIP-1-mediated serine phosphorylation (residue unknown) [49]. Following the formation of the TNFR1 receptor/membrane complex, additional cellular inhibitors of apoptosis 1 and 2 (cIAP1/2) are recruited and bound to TRAF2 [50]. Once bound, cIAP1/2, known formally as E3 ubiquitin ligases, facilitates the ubiquitination of RIP-1 proteins within the same complex [51]. The polyubiquitinated K63-linked chains serve as molecular scaffolding for the IkB [Inhibitor of kB] kinase complex (IKK) [52]. This IKK complex is primarily composed of three domains: IKKα (catalytic), IKKβ (catalytic), and NF-κβ essential modulator (NEMO; regulatory) subunits [53]. At rest, NF-κβ dimers, typically composed of p50 and p65 (RelA), are sequestered in the cytoplasm by inhibitory proteins (IkB) [54]. The activated IKK complex, specifically IKKα, phosphorylates IkBα, the p50/RelA sequestration agent of IkB, along serine residues (S32 and S36) located on the destruction box [55,56]. This culminates in the generation of a phosphodegron motif, which serves as a recognition particle for Beta-Transducing Repeat-containing Protein (βTrCP) [55,56]. βTrCP bound to phosphorylated IkBα orchestrates the polyubiquitination and proteasomal degradation of the IkB subunit [57]. Successively, the liberated p50/RelA dimer freely enters the nucleus and regulates the transcription of antiapoptotic genes [58].

The non-canonical pathway, initiated primarily by TNFR2 receptor activation, does not involve the recruitment of TRADD [59]. Instead, TNFR2 activation leads to the formation of a complex involving TRAF2, TNF receptor-associated factor 3 (TRAF3), and the E3 ubiquitin ligases cIAPs [60]. Once this complex is formed, cIAPs facilitate the ubiquitination of TRAF3 and its subsequent degradation [61]. In its activated form, this complex activates the NF-κB-inducing kinase (NIK), which subsequently phosphorylates and activates IKKα [53]. If there is no receptor activation, TRAF3 remains bound to NIK and promotes its ubiquitination [62]. Activated IKKα then phosphorylates p100, leading to its proteasomal processing into the active form of NF-κβ subunit RelB [63]. The RelB is then dimerized with p52 and subsequently translocated to the nucleus to regulate the transcription of genes, including *bcl-2* and *bcl-xl*, involved in immune cell survival, proliferation, and differentiation [64].

Nuclear translocation of both canonical- and noncanonical-activated dimers enhances the transcription of additional pro-inflammatory mediators, including IL-1β, IL-6, IL-8, and TNF-α [65]. Additionally, NF-κβ activation induces histone acetylation, facilitating the transcription of various genes, including those encoding pro-inflammatory cytokines [66]. When coupled, these pathways elicit an additive effect that continuously exacerbates pro-inflammatory mechanisms in a positive feedback fashion [67].

#### 2.4.2. IL-1 Signaling

The IL-1 family consists of 11 known members, of which, IL-1α, IL-1β, and IL-18 are most notable in MS pathogenesis [68]. The synthesis of IL-1 family cytokines shares several similarities with the synthesis of other cytokines [69]. It is primarily regulated by the binding of antigens to PRRs on innate immune system cells [70]. Upon activation, immune cells initiate signaling cascades that lead to the activation of transcription factors such as NF-κB and activator protein 1 (AP-1) [71]. NF-κβ and AP-1 bind to specific regulatory regions in the IL-1 gene promoter, driving the transcription of IL-1 mRNA [72]. To date, there are two known isoforms of IL-1 (IL-1α and IL-1β) that contribute to inflammatory regulation in MS [73]. Once translated, the inactive, precursory Pro-IL-1 protein undergoes maturation through caspase activation, producing active IL-1 [74].

IL-1 signaling orchestrates a spectrum of pro- and anti-inflammatory tissue responses [75]. The macroscopic outcome of the immune cell response largely hinges on the plasma receptor to which the cytokine binds [76]. IL-1β, much like other cytokine ligands, has a dedicated heterodimeric receptor complex comprised of integral proteins IL-1 Receptor Type 1 (IL-1R1) and IL-1 Receptor Accessory Protein [77] (Figure 2). When IL-1β binds to this complex, the myeloid differentiation primary response 88 (MYD88) adaptor protein is recruited to the cytoplasmic domain of the receptor complex through its Toll/IL-1 receptor (TIR) domain [78]. MYD88 then undergoes homodimerization, which facilitates the formation of a signaling complex [79]. This event is followed by the recruitment of IL-1 Receptor-Associated Kinases 1, 2, and 4 (IRAK1/2/4) [80]. IRAK4, believed to undergo autophosphorylation upon complex binding, then phosphorylates IRAK1 (S_376_/T_387_) and IRAK2 (S_386_/T_399_) [81,82,83]. TNF-α Receptor-Associated Factor 6 (TRAF6) then binds to the phosphorylated IRAK1 and subsequently undergoes a conformational change and undergoes auto-ubiquitination, forming K_63_-linked polyubiquitin chains [84]. Activated TRAF6 serves as a crucial signaling node, mediating downstream signaling events through the activation of various intracellular signaling pathways, including the NF-κB and mitogen-activated protein kinase (MAPK) pathways [85]. Additionally, TRAF6 polyubiquitination can lead to proteasomal degradation in cell-mediated inflammatory responses, though what facilitates the fate of the IL-1β cascade remains unclear [86]. TRAF6 ubiquitination events lead to the recruitment and activation of transforming growth factor-beta-activated kinase 1 (TAK1) and its binding partners, TAK1-Binding Protein 1 (TAB1), TAB2, and TAB3. TAB2 and TAB3 possess specific K_63_-linked ubiquitin chain binding domains, known as ubiquitin-binding in ABIN and NEMO, which facilitate their articulation with the scaffold of polyubiquinated-TRAF6 [87,88]. TAB2 and TAB3 then recruit and activate TAK1 [89]. TAK1, a serine/threonine kinase, subsequently phosphorylates the IKK (S_177_ and S_181_) complex and MAPKs (MKK3; S_189_/T_193_, MKK4; S_257_/T_261_, MKK6; S_207_/T_211_, and MKK7; S_271_/T_275_), leading to the activation of NF-κB and MAPK signaling cascades, respectively [52,90,91]. Activation of these pathways results in the transcriptional regulation of genes involved in inflammatory responses, immune cell activation, and other cellular processes [92]. One such response is the further transcription of successive cytokines, including IL-6 and IL-17 [93,94].

#### 2.4.3. IL-6 Signaling and the JAK/STAT Pathway

IL-6, like IL-1 and TNF-α, is characterized as a pro-inflammatory cytokine [95]. In its classical signaling pathway, IL-6 binds to its heterodimeric receptor complex comprising the IL-6 receptor (IL-6R) and glycoprotein 130 (Gp130) [96]. IL-6R exists in two forms: membrane-bound IL-6R (mIL-6R) and soluble IL-6R (sIL-6R) [97]. Membrane-bound IL-6R is expressed on the surfaces of several cell types, including hepatocytes, leukocytes, and some epithelial cells [98]. Upon binding to IL-6, mIL-6R undergoes conformational changes and forms a complex with the signal-transducing receptor subunit, Gp130 [99]. Gp130 lacks intrinsic kinase activity but is associated with Janus kinases (JAKs), particularly JAK1 and JAK2, which are activated in response to cytokine binding to the receptor complex [100]. Upon activation, JAKs phosphorylate tyrosine residues on the cytoplasmic tail of Gp130, providing docking sites for signal transducer and activator of transcription (STAT) proteins [101]. Once recruited to the receptor complex, STAT proteins become phosphorylated by JAKs along Src homology (SH2) domains and form homo- or hetero-dimers [94]. These STAT dimers translocate to the nucleus, where they regulate the transcription of target genes involved in immune responses, inflammation, cell proliferation, and differentiation [102].

#### 2.4.4. IFN-γ Signaling

IFN-γ signaling is primarily associated with orchestrating inflammation and cell-mediated immune responses [103]. However, recent studies show that IFN-γ may be involved in promoting tumor progression [104]. The production of IFN-γ is tightly regulated by upstream cytokines, notably IL-12 and IL-18, which are primarily secreted by APCs [105]. There are three types of receptors for IFNs that are expressed in nucleated cells [106] (Figure 3). Type I IFN receptors exhibit the broadest specificity, responding to various IFNs, including IFN-α, IFN-β, IFN-τ, IFN-δ, IFN-ɛ, and IFN-ω [107]. Type II IFN receptors, on the other hand, are specifically activated by IFN-γ binding [108]. These receptors exist as tetramers composed of two IFNγR1 and two IFNγR2 subunits. In contrast, type III receptors are highly selective for IFN-λ signaling [109]. Mechanistically, IFN-γ transduces its signals through the JAK/STAT pathway, wherein STAT proteins serve as the primary transcription factors for IFN-γ-induced gene expression [110]. This intricate signaling cascade regulates a wide array of cellular responses, ranging from immune modulation to inflammatory processes, highlighting the diverse roles of IFN-γ in health and disease [111].

#### 2.4.5. MAPK Pathway

In addition to the JAK/STAT pathway, the MAPK pathway is a parallel signaling cascade that plays a crucial role in regulating the expression of inflammatory cytokines [100]. The MAPK pathway is a complex network of signaling proteins, primarily consisting of three main kinases: extracellular signal-regulated kinase (ERK), c-Jun N-terminal kinase, and p38 MAP kinase [112]. The pathway begins with guanine nucleotide exchange factors (GEFs) catalyzing the exchange of GDP for GTP on RAS, a GTPase signaling protein [113]. Upon activation, RAS-GTP recruits and activates rapidly accelerated fibrosarcoma (Raf), which is a kinase [114]. Raf then phosphorylates and activates mitogen-activated protein kinase kinase (MEK; S_218_ and S_222_), which in turn phosphorylates and activates ERK1 (T_202_ and Y_204_) and ERK2 (T_185_ and Y_187_) [115,116,117]. Once activated, ERK1/2 translocates to the nucleus, where it phosphorylates various transcription factors such as Elk-1, c-Fos, and c-Jun [118]. The phosphorylated transcription factors then bind to the responsive elements in the promoter regions of target genes [119], subsequently leading to the transcription of genes encoding proliferative proteins, including cytokines, growth factors, and other regulatory molecules [120].

Among the MAPK pathway kinases, p38 MAPK is particularly relevant in the context of inflammation [121]. It is activated in response to various extracellular stimuli, including pro-inflammatory cytokines and cellular stress [92]. Once activated, p38 MAPK phosphorylates a variety of downstream targets, including ATF-2 and NF-κB, as well as other protein kinases and regulatory proteins involved in inflammatory signaling [122]. This phosphorylation cascade ultimately leads to the expression of inflammatory cytokines including, IL-1β, TNF-α, and IL-6, contributing to the inflammatory response [65].

The MAPK pathway is tightly regulated by various mechanisms, including phosphorylation, ubiquitination, and protein-protein interactions [123]. For instance, negative regulators such as MAPK phosphatases can dephosphorylate and inactivate MAPKs, while scaffold proteins and adaptor molecules facilitate the assembly of signaling complexes and enhance pathway efficiency [124]. Additionally, feedback loops and crosstalk between different signaling pathways further modulate MAPK activity, ensuring precise control of cellular responses to diverse stimuli [125]. Thus, the MAPK pathway serves as a critical mediator of inflammation and represents a promising target for therapeutic intervention in inflammatory diseases, including MS [126].

#### 2.4.6. Sirtuins

Sirtuins represent a family of NAD^+^-dependent protein deacetylases and ADP-ribosyltransferases, prominently implicated in cellular stress, metabolism, and aging [127]. Among them, SIRT1 and SIRT3 have been extensively studied in the context of MS. In MS pathology, dysregulation of sirtuins is evident, notably in their roles in modulating the immune response, oxidative stress, and mitochondrial function [127,128]. SIRT1 exerts anti-inflammatory effects by deacetylating transcription factors, including NF-κB, and inhibiting microglial activation [129]. Conversely, SIRT3 regulates mitochondrial integrity and reactive oxygen species (ROS) detoxification, thereby mitigating neuroinflammation-induced oxidative damage [130]. Dysfunctional sirtuin signaling in MS leads to aberrant immune activation, compromised mitochondrial function, and heightened oxidative stress, which exacerbate neuronal injury, demyelination, and disease progression [130]. Targeting sirtuin pathways holds promise for developing therapeutic strategies aimed at ameliorating neuroinflammation and preserving neuronal integrity in MS [131,132].

### 2.5. Cytokine Dysfunction of MS and Its Effects

In individuals with MS, the levels of TNF-α, IL-1, IL-6, and IFN-γ in the cerebrospinal fluid (CSF) and lesions within the CNS are elevated [133]. Increased production of these cytokines contributes to the activation of macrophages and T cells within the CNS lesions [134]. These immune cells release additional cytokines as part of the inflammatory response, contributing to tissue damage and neuroinflammation [135]. TNF-α-mediated activation of TNFR1 can lead to pro-inflammatory responses, apoptosis, and neurotoxicity, while TNFR2 activation may have neuroprotective and immunoregulatory effects [136]. In MS, there appears to be an imbalance in TNFR1 and TNFR2 signaling, with increased expression of TNFR1 and altered downstream signaling pathways, promoting inflammation and tissue damage [137]. TNF-α has been implicated in the demyelination and neurodegeneration observed in MS [138]. It can directly induce oligodendrocyte apoptosis, leading to demyelination. In addition, TNF-α contributes to neuronal damage through other mechanisms, including excitotoxicity and oxidative stress [139]. Moreover, TNF-α can disrupt BBB integrity by increasing the expression of adhesion molecules on endothelial cells and promoting leukocyte infiltration into the CNS [140]. The disruption of BBB allows immune cells and inflammatory mediators to penetrate the CNS more easily, exacerbating neuroinflammation and tissue damage [141].

The macroscopic outcomes of IL-1 signaling, similar to TNF-α, hinge on the isoform of the IL-1 receptor to which IL-1 family cytokines bind [68]. Given the context of secretory increases in IL-1α and IL-1β expression during MS pathogenesis, understanding imbalanced ligand binding can aid in the development of receptor-antagonistic treatments [142]. While both IL-1α and IL-1β can bind to IL-1R1 on expressive tissues, eliciting a pro-inflammatory response, IL-1β can additionally bind to IL-1R2 [143]. This receptor has been classified as a decoy receptor, which serves to modulate the overproduction of IL-1β in inflamed states [144]. In the context of MS, there exists a dysregulation of the expression of these two receptors. Specifically, IL-1R1, which perpetuates pro-inflammation signaling, has been shown to be overexpressed relative to IL-1R2 [145,146]. Conversely, alterations in IL-1R2 expression or function may affect the regulation of IL-1 activity, further exacerbating inflammatory responses [147].

Regarding IFN-γ receptor dysregulation, the specific imbalance with type II IFNRs in patients with MS remains unclear. However, type II receptor dysregulation has been seen in other autoimmune conditions, such as systemic lupus and rheumatoid arthritis [148,149]. The respective imbalances between TNFR1 and 2 and IL-1R1 and 2 also warrant further investigation as possible therapeutic targets for MS [150].

## 3. Current Treatments for MS

The complexity of the myriad of pathways involved in its pathogenesis demands a multifaceted approach to the management of MS. Current treatment options for MS encompass a diverse array of interventions, including DMTs, symptomatic treatments, acute relapse treatments, rehabilitation therapies, and lifestyle modifications [151]. These treatments are tailored to the specific needs and characteristics of each patient, with the goal of slowing disease progression, minimizing relapses, managing symptoms, and preserving neurological function [152].

### 3.1. Disease-Modifying Therapies (DMT)

DMTs are a cornerstone of treatment for MS, particularly for individuals with relapsing forms of the disease [153]. These medications are designed to modify the immune response, thereby reducing the frequency and severity of relapses and slowing disability progression [154]. DMTs work through various mechanisms, including immunomodulation, anti-inflammatory effects, and modulation of lymphocyte trafficking [155]. Commonly used DMTs include glatiramer acetate (GA), dimethyl fumarate (DMF), teriflunomide, and fingolimod [156]. Treatment selection is based on factors such as disease activity, severity, as well as individual patient characteristics and preferences [157]. Overall, DMTs play a critical role in managing MS by reducing disease activity, delaying progression, and improving long-term outcomes for individuals living with MS [158].

#### 3.1.1. Glatiramer Acetate

Glatiramer acetate (GA) is a DMT that exerts its therapeutic effects through a complex interplay of cellular mechanisms that modulate the immune response and convey neuroprotection [159,160]. GA is a synthetic polypeptide that resembles myelin basic protein, a component of the myelin sheath that is targeted by the autoimmune response [161]. Although the mechanism of action of GA in MS is not fully understood, several ideas have been proposed. One possibility is that it works by inducing immunomodulatory shifts in T cell responses, resulting in the generation of regulatory T cells (Tregs), thus shifting the balance from pro-inflammatory Th1 and Th17 cells to anti-inflammatory Th2 cells [162]. Additionally, GA may act as a decoy antigen, diverting autoreactive T cells away from myelin proteins and towards recognizing GA peptides instead [163]. Furthermore, GA has been shown to activate APCs, such as dendritic cells and macrophages, leading to the release of anti-inflammatory cytokines, such as IL-10, and the suppression of pro-inflammatory cytokines, such as IL-12 and TNF-α [159]. This shift in the cytokine profile contributes to the downregulation of the inflammatory response within the CNS [164]. Finally, GA may exert neuroprotective effects by promoting remyelination, enhancing neuronal survival, and reducing oxidative stress and excitotoxicity [165].

#### 3.1.2. Dimethyl Fumarate

Upon administration, dimethyl fumarate (DMF) is rapidly metabolized to its active form, monomethyl fumarate (MMF), which exerts immunomodulatory and anti-inflammatory effects [166]. One of the primary mechanisms of DMF involves the activation of the nuclear factor erythroid 2-related factor 2 (Nrf2) pathway [167]. MMF activates Nrf2 by covalently modifying cysteine residues on Kelch-like ECH-associated protein 1 (Keap1), a negative regulator of Nrf2 [168]. This modification leads to the dissociation of Nrf2 from Keap1, allowing Nrf2 to translocate to the nuclei, where it binds to antioxidant response elements (AREs) in the promoter regions of genes encoding various antioxidant and cytoprotective proteins [169,170]. Consequently, DMF/MMF upregulates the expression of the genes that code for the following proteins: heme oxygenase-1, NAD(P)H quinone dehydrogenase 1, and glutathione S-transferase [171]. These collectively help to mitigate oxidative stress and maintain cellular redox homeostasis [171]. Additionally, DMF/MMF inhibits the production of IL-17 and IL-23 by modulating the activation of dendritic cells and T cells [172]. DMF/MMF also suppresses the activation and proliferation of various immune cells, including T cells, B cells, and monocytes, through mechanisms that are not fully understood but may involve inhibition of the NF-κB signaling pathway and modulation of mitochondrial function [173,174].

#### 3.1.3. Teriflunomide

As an active metabolite of leflunomide, teriflunomide inhibits dihydroorotate dehydrogenase, a key enzyme involved in pyrimidine synthesis, thereby disrupting the proliferation of activated lymphocytes [175]. By inhibiting pyrimidine synthesis, teriflunomide reduces the proliferation and expansion of autoreactive T and B lymphocytes, which play a central role in the pathogenesis of MS [176]. Furthermore, teriflunomide modulates immune cell function by interfering with the differentiation and activation of various subsets of immune cells [177]. Specifically, teriflunomide inhibits the production of pro-inflammatory cytokines, including IL-17 and IFN-γ, while promoting the secretion of anti-inflammatory cytokines, including IL-10 [175,178]. This shift in the cytokine profile helps to dampen the inflammatory response and restore immune balance within the CNS [179]. Additionally, teriflunomide may exert neuroprotective effects by reducing oxidative stress and mitochondrial dysfunction, which are implicated in the pathogenesis of MS-related neurodegeneration [180].

#### 3.1.4. Fingolimod

Fingolimod acts as a sphingosine-1-phosphate (S1P) receptor modulator [181]. Upon ingestion, it is phosphorylated to fingolimod phosphate (FTY720-P), which resembles S1P [182]. FTY720-P binds to and downregulates S1P1 receptors on lymphocytes, which prevents their egress from lymphoid organs into circulation [183]. This reduces the peripheral pool of circulating lymphocytes available to infiltrate the CNS, thereby mitigating neuroinflammation in MS lesions [183]. Additionally, fingolimod may directly modulate CNS-resident cells, such as astrocytes and microglia, attenuating their activation and pro-inflammatory responses [184]. Beyond its immunomodulatory effects, fingolimod may promote neuroprotective and repair mechanisms, including enhanced neurotrophic factor expression, neuronal survival, and remyelination [184,185].

### 3.2. Symptomatic Treatments and Therapy for MS

Symptomatic treatments and therapies for MS aim to alleviate specific symptoms experienced by individuals living with the condition, thereby improving their quality of life [186]. These interventions target various manifestations of MS, including spasticity, fatigue, pain, bladder dysfunction, and cognitive impairment [187]. Medications like baclofen and tizanidine are commonly prescribed to manage spasticity, while amantadine may be used to address MS-related fatigue [186,188]. Additionally, anticonvulsants or antidepressants may help alleviate neuropathic pain, while medications like oxybutynin or mirabegron can aid in managing bladder dysfunction [189]. Cognitive rehabilitation regimens, including cognitive training and compensatory strategies, are also utilized to address cognitive impairment [190].

### 3.3. Acute Relapse Treatment

The purpose of acute relapse treatment for MS is to alleviate the severity and duration of symptoms associated with flare-ups, known as relapses or exacerbations [191]. These treatments aim to shorten the duration of neurological deficits and facilitate recovery following a relapse [192]. Acute relapse treatments often involve short courses of high-dose corticosteroids, such as intravenous injection of methylprednisolone, which exerts potent anti-inflammatory effects and suppresses immune responses [193]. Corticosteroids reduce inflammation around demyelinated lesions in the CNS, leading to faster resolution of symptoms [194]. In some cases, plasma exchange (also known as plasmapheresis) may be considered for severe relapses that are refractory to corticosteroid therapy [195,196]. Plasma exchange involves removing plasma from the blood and replacing it with a substitute solution, which can effectively remove circulating antibodies and inflammatory mediators implicated in the relapse [197].

### 3.4. Lifestyle Modifications

Lifestyle modifications play a crucial role in managing MS by enhancing overall well-being, minimizing symptoms, and improving quality of life [198]. Regular exercise can help maintain mobility, strength, and flexibility while reducing fatigue and depression, which are commonly associated with MS [199]. A balanced diet rich in fruits, vegetables, lean proteins, and omega-3 fatty acids can support immune function and promote brain health [200]. Adequate hydration is also essential to prevent urinary tract infections and manage the bladder symptoms often experienced by MS patients [201]. Additionally, the implementation of stress management techniques such as mindfulness meditation, yoga, and deep breathing exercises can help alleviate psychological stressors and potentially reduce the risk of MS exacerbations [202]. By incorporating these lifestyle modifications into daily routines, individuals with MS can better manage their symptoms and optimize their overall health and well-being [203].

## 4. Plant-Derived Compounds as Medications for MS

Despite significant advancements in MS, there remains a notable gap in addressing the multifaceted nature of the disease [204]. Current therapeutic strategies primarily target immune dysregulation to mitigate inflammation and slow disease progression [205]. While treatment modalities such as DMTs have demonstrated efficacy in reducing relapse rates and delaying disability accumulation, they often come with limitations such as incomplete efficacy, adverse side effects, and inadequate management of progressive forms of MS [206]. Furthermore, there is a growing recognition of the need for alternative treatment modalities that can offer greater efficacy, tolerability, and neuroprotective effects [207]. In this context, plant-derived compounds present a promising avenue for addressing these unmet needs in MS management [208]. With their diverse array of bioactivities, plant-derived medications have the potential to complement existing therapies by targeting inflammation, oxidative stress, and neurodegeneration in MS [209]. By harnessing the therapeutic properties of phytochemicals, such as phenylpropanoids, alkaloids, and terpenoids, there is an opportunity to fill some of the gaps in currently available treatment options for MS and to improve outcomes for individuals living with this complex autoimmune disorder [11].

### 4.1. Polyphenols 

#### 4.1.1. Epigallocatechin-3-Gallate 

Epigallocatechin-3-gallate (EGCG) (Figure 4A), a potent polyphenol abundant in green tea, has garnered considerable interest for its wide-ranging health benefits [210,211]. Derived primarily from the leaves of the *Camellia sinensis* plant, EGCG belongs to the catechin group of compounds [212]. While green tea serves as the primary dietary source of EGCG, smaller quantities can also be found in white tea and oolong tea [213]. EGCG has a bioavailability of 0.1% (Table 1) [214], and is available in concentrated supplement form for those seeking targeted consumption [215].

Recent research highlights the ability of EGCG to inhibit the effects of several pro-inflammatory cytokines, including TNF-α, IL-1β, and IL-6, by targeting key signaling pathways [222,223]. By blocking the IKK complex and preventing the phosphorylation of IkB, EGCG inhibits the release of NF-κB, a critical mediator of inflammation [224]. Moreover, EGCG disrupts the MAPK signaling pathway, further reducing IL-1β production [225]. This dual mechanism leads to a decrease in pro-inflammatory cytokine levels while promoting the production of the anti-inflammatory cytokines, IL-10 and TGF-β [179,226].

Beyond its immunomodulatory effects, EGCG demonstrates neuroprotective properties, shielding neurons from oxidative stress and apoptosis and potentially promoting remyelination [227]. The accumulation of EGCG within the mitochondria may protect against neuronal damage by reducing induced apoptosis [228]. Additionally, EGCG has shown promising results in enhancing cell viability, reducing markers of stress and apoptosis, and protecting against various forms of toxicity [229]. These protective effects extend to mitigating glutamate excitotoxicity and preserving mitochondrial function, ultimately enhancing cognitive function and prolonging lifespan [230]. Outside the realm of neurology, EGCG has shown promise in cancer treatment, cardiovascular health, weight management, diabetes management, and skin health, underscoring its versatility and potential as a multifaceted therapeutic agent [229,231].

In an 18-month clinical trial, although a dose of 800 mg EGCG was shown to be safe and bioavailable in patients with relapsing-remitting MS, it had no additional effect on the GA treatment on the MRI or immune parameters [232]. However, the same dose of EGCG combined with 60 mL of coconut oil showed a significant improvement in some gait parameters and balance in MS patients over a 4-month nutritional intervention study, which may suggest a neuroprotective effect [233]. From the same study, the combined effect of EDCG, coconut oil, and a Mediterranean isocaloric diet, a promising protective effect against cardiac risk by improving levels of albumin, beta-hydroxybutarate, and paraoxonase 1 and anthropometric parameters such as waist-to-hip ratio and muscle mass [234]. In an experimental study, different doses of EGCG had an anti-fatigue effect by improving associated blood parameters and increasing glycogen content in the liver and the muscles of the mice [235]. These findings suggest that, when combined with the appropriate nutritional intervention, EGCG is a promising polyphenol for the management of MS symptoms. However, more clinical research is needed to confirm this.

#### 4.1.2. Resveratrol

Resveratrol (Figure 4B), a natural polyphenol abundant in plants such as grapes and berries, has emerged as a promising compound in the realm of MS research due to its notable anti-inflammatory properties [236]. Studies have revealed, similarly to EGCG, its capacity to target key inflammatory cytokines implicated in MS pathogenesis, including TNF-α, IL-1β, IL-6, IL-17, and IFN-γ, effectively dampening the inflammatory cascade [226,237]. Moreover, the neuroprotective effects of resveratrol and its ability to promote remyelination further underscore its potential in attenuating MS progression [238]. These beneficial effects are attributed to its modulation of various signaling pathways, such as suppressing NF-κB and MAPK while activating sirtuins, which collectively reduce inflammation and neurodegeneration [239,240]. The oral bioavailability of resveratrol is low, <1% (Table 1) [241]. However, in a mouse model of MS, resveratrol-loaded macrophage exosomes administered intranasally reduced inflammatory parameters in the CNS and relived disease progression via microglia targeting [242]. As such, resveratrol stands as a promising candidate for the development of novel therapeutic interventions against MS, although further research is necessary to delineate its precise mechanisms of action and therapeutic efficacy [243].

#### 4.1.3. Quercetin

Quercetin (Figure 4C), a natural flavanol found widely in fruits, vegetables, and grains, exhibits numerous pharmacological effects that make it a promising candidate for neuroprotection and multiple sclerosis (MS) treatment. It reportedly has a bioavailability of 16% (Table 1) [244], which is surprisingly higher relative to most polyphenols. Its therapeutic potential is attributed to its ability to modulate key signaling pathways involved in oxidative stress and inflammation, particularly the nuclear factor erythroid 2–related factor 2 (Nrf2) and heme oxygenase-1 (HO-1) pathways [245]. Quercetin activates Sirtuin 1 (SIRT1), which has been linked to neuroprotection and anti-aging effects [246]. Quercetin also stimulates autophagy in Schwann cells, enhancing their ability to cope with neurodegenerative stress [247]. The antioxidative and anti-apoptotic activities of quercetin contribute to reducing hypoxia-induced memory dysfunction and increasing neuronal survival [248,249].

With respect to MS, quercetin impacts both demyelination and remyelination [250,251]. Quercetin also modulates inflammatory responses by inhibiting key cytokines such as TNF-α, IL-1β, and IL-6, inhibiting dendritic cells and Th17 cells, and shifting microglial activation to a neuroprotective M2 phenotype [252,253,254]. Inhibition of Th17 cell differentiation is achieved by targeting STAT4 [255,256]. These effects underscore the potential of quercetin as a complementary MS treatment, addressing both inflammation and neurodegeneration.

In an ethidium bromide-induced demyelination rat model, quercetin treatment (50 mg/kg/day) prevented additional demyelination, improved remyelination, enhanced locomotor activity, inhibited lipid peroxidation, and preserved acetylcholinesterase (AChE) activity [257]. A similar model confirmed that quercetin protected Na^+^/K^+^-ATPase function in both demyelination and remyelination phases, decreased oxidative stress, and maintained AChE activity [258]. Additionally, in a lysolecithin-induced demyelination model in the optic chiasm, quercetin treatment led to reduced visual evoked potential latency, diminished demyelination, and enhanced remyelination [259]. In experimental allergic EAE models, quercetin reduced disease progression by controlling myeloperoxidase activity, nitric oxide levels, and lipid peroxidation [255,256]. It also inhibited IL-12-induced T cell proliferation and Th1 differentiation. In vitro studies further highlight the ability of quercetin to decrease cytokine levels, such as IL-1β and TNF-α, and modulate matrix metalloproteinases (MMPs) in peripheral blood mononuclear cells from MS patients [253]. Quercetin-loaded nanoparticles have also shown the potential to reduce demyelination and inflammation in preclinical models [260].

#### 4.1.4. Ellagic Acid

Ellagic acid (Figure 4D), a polyphenol abundant in the Mediterranean diet, despite having a low bioavailability of <0.2% (Table 1) [261], shows significant promise for MS treatment [262] as it attenuates demyelination and neuroinflammation in MS models [263,264]. The anti-inflammatory effects of ellagic acid are partly mediated through its ability to inhibit NF-κB signaling [265,266]. By suppressing NF-κB activation, ellagic acid reduces the production of pro-inflammatory cytokines, including TNF-α, IL-1β, and IL-6, which are known to exacerbate neuroinflammation in MS [263]. In the MOG35−55-immunized EAE model, high-dose ellagic acid (50 mg/kg) alleviates clinical symptoms, improves motor function, and reduces neurological deficits [263]. It counteracts astrogliosis, astrocyte activation, demyelination, and axonal loss, attenuating neuroinflammation and axonal damage by modulating the NLRP3 inflammasome and pyroptotic pathways, reducing pro-inflammatory cytokines, and increasing IL-4 levels and GATA3 expression [264,265]. Ellagic acid has been shown to inhibit apoptosis in neural cells by modulating various signaling pathways, including the phosphatidylinositol 3-kinase/protein kinase B (PI3K/Akt) pathway. By enhancing PI3K/Akt signaling, ellagic acid promotes cell survival and inhibits the apoptotic pathways that are often triggered during neurodegeneration [267,268].

Human studies with ellagic acid supplementation (90 mg twice daily for 12 weeks) show improvements in health markers, including reduced BDI-II scores (depression index score), IFN-γ, NO, cortisol, and IDO gene expression, and increased brain-derived neurotrophic factor and serotonin levels [269,270]. In a cuprizone-induced demyelination model, ellagic acid ameliorates behavioral impairments and counters oxidative stress by enhancing antioxidant enzyme activities [271]. The compound also positively impacts gut microbiota, promoting beneficial bacteria and increasing propionate levels, which correlate with reduced EAE symptoms [262]. These findings highlight the role of ellagic acid in modulating immune responses and improving neurological health, thus offering a promising therapeutic avenue for MS.

#### 4.1.5. Luteolin

Luteolin (Figure 4E) has demonstrated significant potential as a therapeutic agent for MS by modulating key pathways involved in remyelination and inflammation [272,273]. It has been shown to have a bioavailability of 4.1% (Table 1) [274]. In rodent models of MS, luteolin effectively inhibits the Nrf2 pathway in astrocytes, which is crucial for cholesterol biosynthesis and transfer to oligodendrocytes. This process is essential for myelin repair, as sustained Nrf2 activation impairs oligodendrocyte survival and remyelination. Luteolin’s inhibition of Nrf2 restores cholesterol biosynthesis and supports oligodendrocyte function, thereby promoting remyelination [275]. This mechanism highlights a novel therapeutic target in MS, where luteolin can modulate astrocyte-oligodendrocyte interactions to enhance central nervous system regeneration.

Furthermore, luteolin exhibits robust anti-inflammatory effects, which are critical in the context of MS pathology. It significantly reduces the production of pro-inflammatory cytokines, including IL-1β and TNF-α, and inhibits STAT3 phosphorylation, a key mediator in T cell activation [273,276,277]. This dual-action mechanism not only mitigates neuroinflammation but also supports oligodendrocyte survival and remyelination, positioning luteolin as a promising candidate for the treatment of MS.

#### 4.1.6. Curcumin

Curcumin (Figure 4F), derived from turmeric, has a bioavailability of 60–66%, as reported in one study (Table 1) [278]. However, the circulating concentrations of curcumin after oral administration are low, with its various metabolites being primarily detected. Regardless, curcumin has demonstrated potential in enhancing IFN β-1a therapy for MS by improving radiological inflammation markers [279,280]. In EAE models, polymerized nano-curcumin (PNC) significantly reduced disease scores and symptoms like paralysis and motor deficits [281,282]. PNC modulates immune responses by lowering pro-inflammatory factors and raising anti-inflammatory cytokines such as IL-4, IL-10, and TGF-β [283,284]. It also boosts FOXP3 expression, a key transcription factor for regulatory T cells, and enhances HO-1 expression via the Nrf2 pathway, thereby reducing neuroinflammation [285,286].

The effects of curcumin extend to promoting myelin repair and reducing glial activation [282,287]. Curcumin-loaded nanoparticles show greater efficacy than free curcumin in decreasing immune cell infiltration and demyelination in the corpus callosum [287]. Clinical studies reveal benefits in reducing Expanded Disability Status Scale (EDSS) scores and increasing anti-inflammatory markers like TGF-β and IL-10 [285,286]. Although curcumin combined with IFN β-1a did not significantly alter EDSS scores, it improved the anti-inflammatory effects of IFN β-1a without increasing adverse reactions. Curcumin also influences gut microbiota, which affects neuroinflammation and disease progression, highlighting its potential as an adjunct therapy for MS [288].

### 4.2. Alkaloids

#### 4.2.1. Caffeine

Caffeine (Figure 5A) is a purine alkaloid, which is subcategorized as a trimethylxanthine, a member of the methylxanthine class of pharmacologic agents [289,290]. Caffeine has a bioavailability of 99% (Table 2) [291], and prominent sources include coffee, tea, soda, and energy drinks [290]. Extensive research has demonstrated the ability of caffeine to protect against Alzheimer’s and Parkinson’s disease, to oppose oxidative stress and inflammation, and to act as a bronchodilator and vasodilator [292,293,294]. Relevant to MS, caffeine suppresses inflammation via inhibiting NF-κB [295]. Specifically, it disrupts the NF-κB signaling progression by blocking the translocation of p50 and p65 subunits into the nuclei [296]. Inhibition of NF-κB by caffeine also reduces NLRP3 inflammasome, which contributes to NF-κB-stimulated transcription of IL-1β and IL-18 [295,297]. Caffeine may also prevent inflammation by down-regulating the NLRC4 inflammasome [298], which is the main protease responsible for converting pro-IL-1β into active mature IL-1β. Thus, caffeine can reduce IL-1β production by inhibiting NF-κB, NLRP3, and NLRC4.

While it is not fully elucidated as to how caffeine suppresses inflammation, it may modulate immune responses by acting as a non-specific antagonist of adenosine receptors (ARs), which are the main targets of ingested caffeine. There are four known subtypes of ARs, A_1_, A_2A_, A_2B_, and A_3_, all of which are coupled to G-proteins. However, each subtype of ARs has a distinct pharmacological profile, tissue distribution, and G-protein coupling [306]. Therefore, adenosine, the endogenous ligand for ARs, can elicit many physiological or pathological effects by acting on these receptors in a dose-dependent manner. Accordingly, caffeine exerts various effects depending on the concentration. Low doses of caffeine can stimulate cAMP production by blocking the A_1_ receptor, which, along with A_3_, is coupled to Gi/Go protein to inhibit adenylate cyclase (AC) activity [307]. Subsequently, increased cAMP concentrations suppress proinflammatory cytokine production partially via activating the repressor transcription factor CCAAT displacement protein (CDP) [308]. Cyclic AMP also directly binds to NLRP3, thus directly inhibiting the assembly of the inflammasome [309]. Further, caffeine has been shown to inhibit phosphodiesterases, which hydrolyze cAMP to AMP [310]. The A_2_ receptor is coupled to Gαs, and its activation stimulates AC activity and cAMP production. At high concentrations, caffeine antagonizes the A_2_ receptor, thereby inhibiting cAMP formation, leading to the increased production of proinflammatory cytokines [307]. However, the circulating concentrations of caffeine required for inhibiting the A_2_ receptor in vivo may not be achievable through dietary intake of coffee or caffeine. It was found that sufficient antagonism of A_2_ requires caffeine concentrations ≥ 100 µM [311]. Orally ingested caffeine typically reaches a peak plasma concentration (Cmax) between 15 and 120 min in healthy adults [291]. A recent study shows that oral administration of 200 mg of caffeine (roughly the amount of caffeine found in two 240 mL servings of coffee) in healthy men resulted in a Cmax of 3.4 mg/L, which equates to a plasma concentration of 17.51 µM [312]. To achieve the plasma concentrations of caffeine sufficient to inhibit A_2_ receptor and cAMP production would require ~1100 mg of orally ingested caffeine, which is about 2.75 times the USDA-published safe limit of 400 mg/day [313]. Thus, moderate caffeine consumption is unlikely to cause the A_2_ antagonism-mediated inhibition of cAMP production. Although further investigation is needed to more thoroughly elucidate the cellular mechanisms behind its anti-inflammatory effects, caffeine presents a promising avenue for future MS therapies.

#### 4.2.2. Harmane

Harmane (Figure 5B), a β-carboline alkaloid, exhibits potential as an anti-inflammatory agent in MS treatment [314]. The compound is derived from the *Peganum harmala* plant. It has a bioavailability of 19% (Table 2) [315], and can be naturally found in several foods and beverages, including soy sauce, toasted bread, barley, coffee, and fermented alcohol-containing beverages [314,315,316,317]. Additionally, harmane can be found in many cooked meats such as beef, mutton, and chicken, with greater concentrations of harmane being found in meats that have been cooked for longer periods of time and at higher temperatures [314]. This is because harmane can be formed through the Maillard reaction, a process often used in food processing to imbue food products with appealing flavors and colors that involves interactions between the free amine groups of proteins and the carbonyl groups of carbohydrates [318].

Harmane is one of the alkaloids present in *Peganum harmala*, which was often used as a medicinal herb in ancient times in certain regions of the world to treat multiple diseases, including various cancers [319]. More recently, research demonstrated that harmane can counteract inflammation, primarily by inhibiting myeloperoxidase (MPO) [320]. MPO is expressed most abundantly in neutrophils and aids in defense against pathogens by catalyzing the formation of many ROS, including hypochlorous acid (HOCl), which is a potent antimicrobial agent and one of the strongest oxidant molecules produced in the human body [321]. In contrast to their beneficial contributions to the efficacy of the immune system, the ROS produced by MPO have also been implicated in a wide variety of cardiovascular, neurodegenerative, and autoimmune diseases [322]. In the context of MS, MPO causes the formation of ROS such as hypochlorous acid, tyrosyl radicals, and aldehydes, which can increase the production of proinflammatory cytokines (IL-1α and TNF-α, for example) [323]. ROS can activate the NF-κB pathway and subsequently increase IL-1 and TNF-α expression [324]. Interestingly, MPO inhibitors have been shown to decrease the expression of the proinflammatory cytokines IL-1α and TNF-α [325]. Thus, harmane, as an MPO inhibitor, might be useful in attenuating MS pathogenesis.

Although the current literature is limited with respect to the effects of harmane supplementation on the pathogenesis of MS in vivo, one study sought to determine the neuroactive effects of β-carbolines supplied as pure compounds versus a natural source (coffee substitute) in a murine model [326]. The results of the study indicated that the animal diets enriched with coffee substitutes resulted in a higher concentration of harmane in the blood and had a positive effect on animal activity [326]. Further, the lack of significant differences between the health parameters of the control and experimental groups suggested that there was no negative effect on the general health of the animals associated with the addition of harmane to the diet [326].

#### 4.2.3. Trigonelline

Trigonelline (Figure 5C) is a pyridine alkaloid derived from nicotinic acid and is classified as a methylnicotinic acid [327]. Dietary sources of trigonelline include fenugreek seeds, garden peas, hemp seeds, oats, coffee, and coffee byproducts [328,329]. Trigonelline has a bioavailability of 64.42% (Table 2) [329] and has demonstrated promise as an anti-oxidative, anti-hyperglycemic, anti-hyperlipidemic, anti-hypercholesterolemic, anti-cariogenic, and anti-microbial agent [329]. In addition, trigonelline had neuroprotective effects in animal models of diabetes, Alzheimer’s disease, Parkinson’s disease, and modulated process that involves nervous system development and inflammation [330]. Specifically, regarding inflammation, trigonelline prevents the transcriptional upregulation of the p50 and p65 subunits of NF-κB [331,332], thereby reducing the expression of TNF-α, IL-1β, IL-6 [65], suggesting that it may potentially be used to prevent the pathogenesis of MS.

While trigonelline has yet to be thoroughly investigated for its effects on MS, studies evaluating its efficacy in the treatment of other neurodegenerative diseases evidence the value of trigonelline as a prospective compound for future investigation in the context of MS. One such study demonstrated that trigonelline significantly mitigated oxidative stress in LPS-treated mice by increasing the levels of antioxidant defense enzymes and decreasing the lipid peroxidation [333]. The study further showed that TNF-α and IL-6 levels, which had been significantly elevated in the mice after LPS administration, were significantly reduced following trigonelline administration at doses of 50 and 100 mg/kg [333].

Another study sought to evaluate the neuroprotective effects of trigonelline by examining the ability of the compound to restore amyloid β (Aβ)-induced axonal degeneration and improve memory function in Alzheimer’s disease 5XFAD model mice [334]. The results demonstrated that oral administration of trigonelline to 5XFAD mice for 14 days resulted in significantly improved object recognition memory and object location memory and normalized neurofilament light levels in the cerebral cortex, which is a biomarker of axonal damage [334].

### 4.3. Terpenoids

#### 4.3.1. Cafestol

Cafestol (Figure 5D) is a fat-soluble ent-kaurene diterpenoid that is derived from the beans of the *Coffea arabica* plant and has been the subject of numerous pharmacological studies due to its various beneficial biological activities, including anti-inflammatory, anti-carcinogenic, anti-angiogenic, anti-diabetic, anti-oxidant [335,336,337] and neuroprotective [338] effects. The concentration of cafestol found in a cup of coffee can vary greatly depending on the quality, blend, and method of preparation. Unfiltered coffee has been shown to contain significantly larger concentrations of cafestol than filtered coffee [339]. The majority (64–70%) of cafestol is absorbed by the duodenum of healthy individuals (Table 2) [340].

Emerging evidence has shown that cafestol targets several biological pathways to exert its anti-inflammatory effects. This is primarily accomplished by regulating chemokines intercellular adhesion molecule-1 (ICAM1), monocyte chemoattractant protein-1 (MCP1), and IL8 [336,341]. It was shown that cafestol can inhibit the secretion of inflammatory mediators induced by cyclic strain in human umbilical vein endothelial cells (HUVECs) [341]. It is believed that cafestol attenuates ROS production, which in turn prevents MAPK phosphorylation, leading to the reduction in the production of these inflammatory mediators [341]. In addition, cafestol has been demonstrated to significantly reduce TNF-α and IL-1β levels and inhibit cardiac apoptosis by modulating tissue levels of Bax and Caspase-3 [342]. It has further been reported that cafestol can effectively block the AP-1 pathway by directly inhibiting the activity of ERK2 and consequently reducing the production of PEG2 and its associated pro-inflammatory activities [343].

Although the existing literature contains an abundance of in vitro studies investigating the biological activities of cafestol, the available in vivo studies pertaining to MS are limited due to greater emphasis being placed on its remarkable anti-diabetic effects. Although further research is needed, the affordability and abundance of cafestol, as well as its numerous benefits and minimal side effects, make it a promising option for future investigation in the context of MS.

#### 4.3.2. Ursolic Acid

Ursolic acid (UA) (Figure 5E) is a pentacyclic triterpenoid and a secondary metabolite present in many commonly used plants, including fruits, vegetables, and herbs such as thyme, rosemary, lavender, oregano, and mint [344]. Despite its very low bioavailability (0.03%) (Table 2) [345], the anti-inflammatory and neuroprotective effects of UA make it a promising candidate with respect to the development of novel therapies for MS. Recent research has demonstrated that the administration of UA can significantly increase the expression of the anti-inflammatory cytokine IL-10 [346]. Additionally, UA has been shown to attenuate amyloid β (Aβ)-induced memory impairments through amelioration of oxidative stress and downregulation of IL-1β, IL-6, and TNF-α levels in the hippocampus of mice [347]. Finally, it was found that UA markedly inhibited LPS-induced IκBα phosphorylation and degradation, NF-κB p65 nuclear translocation, and p38 activation in the mouse brain but did not affect the activation of TLR4, MyD88, ERK, JNK, and Akt [348]. These data suggest that UA may hold the potential to mitigate inflammation-associated brain disorders by blocking the p38 and NF-κB signaling pathways and inhibiting the production of pro-inflammatory factors.

In addition to its anti-inflammatory properties, UA has also been shown to exert neuroprotective effects and actively combat demyelination in the central nervous system [349]. The results of a recent study indicated that treatment with UA increased the number of new oligodendrocyte lineage cells and myelination by reducing inflammation and preventing gliosis [349]. Although the study was primarily concerned with the investigation of UA in the context of Parkinson’s disease, microglial activation and proliferation are key features of MS pathology as well. Given its role as an anti-inflammatory mediator and its ability to preserve myelin and actively promote the remyelination of axons, UA holds immense therapeutic potential for treating MS.

#### 4.3.3. Celestrol

Celastrol (Figure 5F), a pentacyclic triterpene derived from the root of the *Tripterygium wilfordii* plant, has shown significant therapeutic potential across various conditions, including diabetes, metabolic dysfunction, irritable bowel syndrome, and Alzheimer’s disease [350,351,352,353]. Its chemical structure is defined as 3-hydroxy-9β,13α-dimethyl-2-oxo-24,25,26-trinoroleana-1(10),3,5,7-tetraen-29-oic acid [350] and it has a bioavailability of 17% (Table 2) [354]. The efficacy of celestrol has been demonstrated in numerous human clinical trials, and it has been recognized for its ability to restore lipid metabolism and modulate protein homeostasis.

In MS research, celastrol exhibits potent anti-inflammatory and neuroprotective properties. Celastrol mechanistically inhibits pro-inflammatory cytokines, including IFN-γ and IL-17, while upregulating anti-inflammatory cytokines, such as IL-4 [355]. This modulation is achieved by blocking key transcription factors such as STAT3 and retinoid-related orphan receptor gamma t, which are involved in Th17 cell differentiation and inflammation [356]. The action of celestrol extends to the inhibition of NF-κB and AP-1, crucial transcription factors that drive inflammation [355,357]. It also inhibits LPS-induced production of IL-1β, TNF-α, and IL-6, potentially through cyclooxygenase-2 inhibition [358,359]. Additionally, celastrol promotes mitochondrial autophagy by acting as a Nur77 ligand, which contributes to its neuroprotective effects [360].

In MS models, the effects of celastrol are particularly pronounced in the spinal cord and optic nerve [355]. It significantly reduces neuroinflammation and apoptosis, as evidenced by lower levels of nitrites, reduced immunohistochemical expression of TLR2 and CD3+ T-lymphocytes, and improvements in histopathological scores [355]. The treatment also decreases levels of chemokines (including RANTES, MCP-1, MIP-1α, and GRO/KC) and cytokines (including TNF-α and IL-1β) [361]. In the optic nerve, Celastrol mitigates severe inflammatory responses and microgliosis, and it restores apoptotic balance. In EAE models of MS, celastrol reduces neurobehavioral abnormalities, inflammatory infiltration, and demyelination. It downregulates the pro-inflammatory cytokines IFN-γ and IL-17 while upregulating anti-inflammatory cytokines, including IL-4 [355]. Celastrol also mitigates severe inflammatory responses and microgliosis, reduces chemokines, and inhibits IL-17 expression [356].

Despite its promising preclinical results, the use of celestrol is associated with potential risks, including microglia cytotoxicity at elevated concentrations (100–1000 nM), and chronic use is associated with heart and liver damage [362,363,364].

### 4.4. Catechol

#### Hydroxytyrosol

Hydroxytyrosol (HT) (Figure 5G) is a potent phenolic compound primarily found in olive oil and olive leaves and is known for its robust antioxidant properties [365]. The bioavailability of HT is 75% when administered in an aqueous solution (Table 2) [366] and its amphipathic nature facilitates effective absorption and distribution [367]. HT primarily scavenges reactive oxygen species (ROS) both intracellularly and extracellularly, addressing oxidative stress associated with neurodegenerative diseases [368,369]. The ability of HT to cross the blood-brain barrier (BBB) is particularly relevant for treating CNS disorders, including MS [370,371].

In MS, HT counteracts chronic inflammation and oxidative stress that contribute to myelin degradation and neuronal damage [372]. By reducing the expression and activity of MMP-9 and MMP-2, HT helps maintain BBB integrity and limits immune cell infiltration into the CNS [373,374,375]. HT also diminishes oxidative stress and lipid peroxidation in MS, enhances antioxidant enzyme activity such as glutathione peroxidase, and regulates iron metabolism [374]. These actions collectively support the potential of HT as a treatment for alleviating symptoms and slowing the progression of MS.

## 5. Conclusions

The emerging evidence presented in this review underscores the promising potential of plant-derived compounds as adjunctive therapies for MS [376]. With their diverse array of bioactive compounds and multifaceted mechanisms of action, including anti-inflammatory and neuroprotective properties, plant-derived compounds offer a complementary approach to existing treatments [377]. By targeting inflammation, oxidative stress, and neurodegeneration, these compounds address critical aspects of MS pathogenesis. Moreover, their ability to modulate immune responses and promote neuroregeneration suggests a broader therapeutic scope beyond symptom management. However, further research is needed to elucidate the specific mechanisms of action, optimize dosing regimens, and evaluate long-term efficacy and safety profiles. By harnessing the therapeutic potential of phytochemicals, such as alkaloids, phenylpropanoids, and terpenoids, there is an opportunity to enhance the clinical management of MS and improve outcomes for affected individuals. Ultimately, the integration of plant-derived pharmacologic agents into the MS treatment paradigm holds promise for addressing the complex and multifaceted nature of this autoimmune disorder, offering hope for improved quality of life and functional outcomes for patients.

## Figures and Tables

**Figure 1 nutrients-16-02996-f001:**
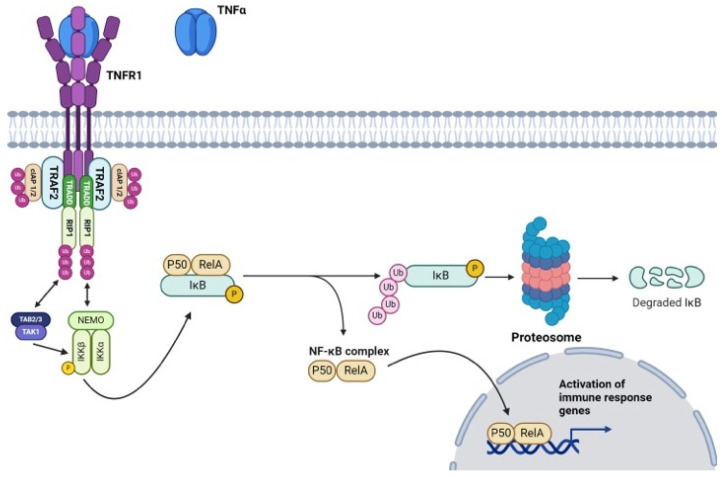
The canonical TNF-α signaling pathway activates NF-κB through TNF receptor binding, involving TRADD, RIP-1, and the IKK complex, leading to the transcription of various genes that are involved in the immune response (Created with https://www.biorender.com/ accessed on 3 July 2024).

**Figure 2 nutrients-16-02996-f002:**
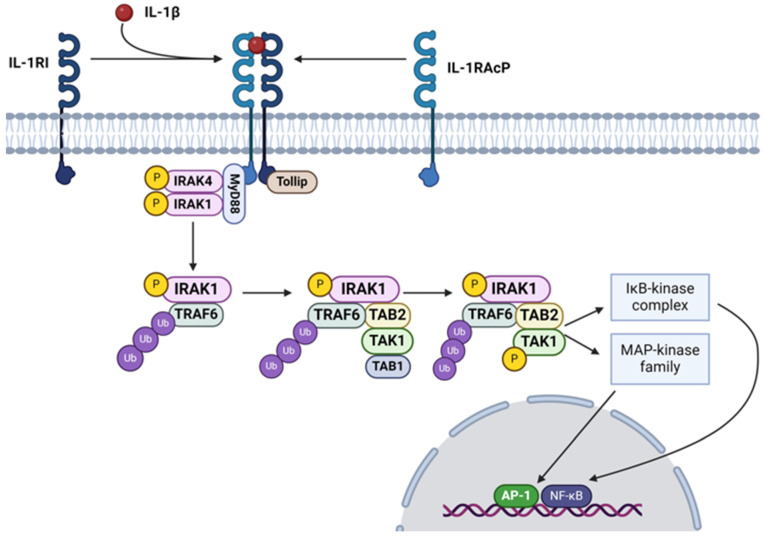
The IL-1β signaling pathway activates NF-κB and MAPKs through the IL-1 receptor, leading to the transcription of genes involved in inflammation and immune responses (Created with https://www.biorender.com/ accessed on 3 July 2024).

**Figure 3 nutrients-16-02996-f003:**
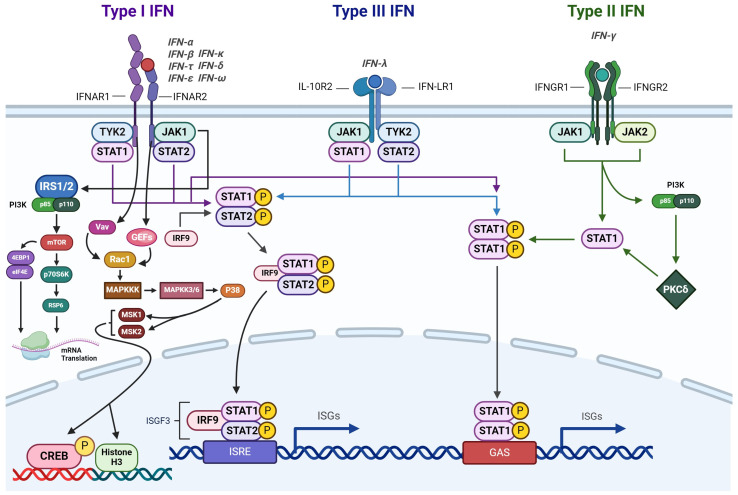
IFN signaling activates the JAK-STAT pathway, leading to the transcription of genes involved in the immune response (Created with https://www.biorender.com/ accessed on 3 July 2024).

**Figure 4 nutrients-16-02996-f004:**
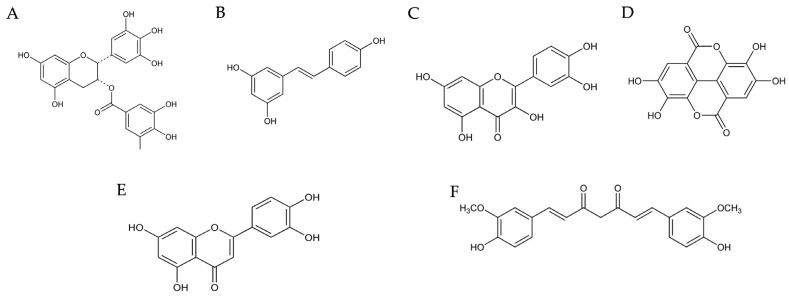
Structures of the polyphenols described in this article. EGCG (**A**), resveratrol (**B**), quercetin (**C**), ellagic acid (**D**), luteolin (**E**), and curcumin (**F**) [216,217,218,219,220,221].

**Figure 5 nutrients-16-02996-f005:**
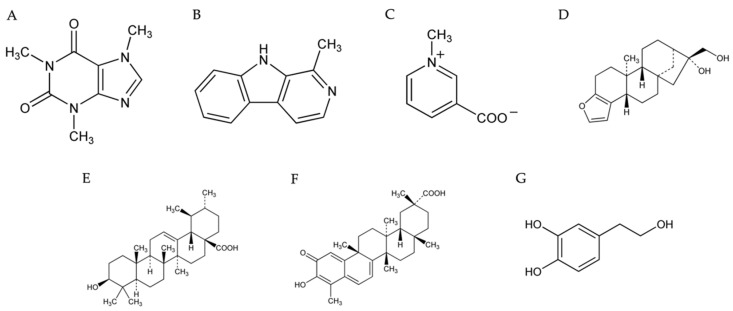
Chemical structures of the alkaloids, terpenes, and catechol described in this article. Caffeine (**A**), harmane (**B**), trigonelline (**C**), cafestol (**D**), ursolic acid (**E**), celastrol (**F**), and hydroxytyrosol (**G**) [299,300,301,302,303,304,305].

**Table 1 nutrients-16-02996-t001:** Summary of bioavailability of each polyphenol discussed with the potential to treat MS-associated symptoms. (#) denotes an in vivo study and (^) a clinical study.

Compound	EGCG	Resveratrol	Quercetin	Ellagic Acid	Luteolin	Curcumin
Bioavailability	0.1% ^^^	<1% ^^^	16% ^#^	0.2% ^#^	4.1% ^#^	60–66% ^#^

**Table 2 nutrients-16-02996-t002:** Summary of bioavailability of each alkaloid, terpene, and catechol discussed with the potential to treat MS-associated symptoms. (#) denotes an in vivo study and (^) a clinical study.

Compound	Caffeine	Harmane	Trigonelline	Cafestol	UA	Celastrol	Hydroxytyrosol
Bioavailability	99% ^^^	19% ^#^	64.42% ^#^	3–5% ^^^	90% ^^^	17.06% ^#^	75% ^#^

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
