# Peer review of "Potential Application of Plant-Derived Compounds in Multiple Sclerosis Management"

_nutrients, 2024, doi:10.3390/nu16172996_

Round 1

Reviewer 1 Report

Comments and Suggestions for Authors

The following corretions are suggested in order to improve the quality of the manuscript:

-please include the structures of the phytochemicals;

-please include considerations about their bioavailability that is crucial for any compound, especially for the polyphenols that are affected by low oral bioavailability;

-please consider to improve the sections alkaloids and terpenoids. The potential applications of the phytochemcals on MS is based in most cases on in vitro studies not directly related to MS.

Comments on the Quality of English Language

Minor corrections

Author Response

Comments 1: please include the structures of the phytochemicals;

Thank you for your time reviewing and critiquing the manuscript. We have carefully addressed all comments and suggested revisions, as shown point by point below:

Comments 1: please include the structures of the phytochemicals.

Response: The structures of the phytochemicals, including those newly added, have been included as two multi-panel figures (Figure 4,5). Each panel is cited at the first mention of the compound in section 4.

Comments 2: please include considerations about their bioavailability that is crucial for any compound, especially for the polyphenols that are affected by low oral bioavailability;

Response: We agree with this comment and have included those values in the text and as summative tables for easy reference. The changes are highlighted in red.

Comments 3: please consider to improve the sections alkaloids and terpenoids. The potential applications of the phytochemcals on MS is based in most cases on in vitro studies not directly related to MS.

Response: Thank you for this suggestion. We have added our review of literature from animal studies and clarified, where it might have been otherwise ambiguous, the model used in the cited studies. The changes are highlighted in red (Lines 726-728; 739-745;751-755;762-770; 784-799; 821-823; 841-845; 851-853).  We have also added review of quercetin, ellagic acid, luteolin, curcumin (Pages 13-16), celastrol (pages 19-20), and hydroxytyrosol (pages 20-21), all of which have shown promise in human clinical trials, especially curcumin, celastrol, and ellagic acid.

Comments 4: on the Quality of English Language: Minor corrections.

Response: We have made spelling and grammatical corrections.

Reviewer 2 Report

Comments and Suggestions for Authors

In the manuscript presented for review, the authors conducted a literature review on the possibility of using substances of plant origin in the treatment and prevention of sclerosis. I think the idea for a review article is good and this type of study will certainly be helpful to people dealing with this issue. The entire manuscript has been divided into two parts. In the first part, the authors presented the mechanisms responsible for the development of sclerosis. This was presented in more detail together with graphic models of the mechanisms. In the second part, the authors began to describe the role of substances of plant origin in preventing the occurrence of sclerosis. While the first part is strong in my opinion, the rest of the manuscript is disappointing. The reader would expect more from the title. The authors limit themselves to only a few selected compounds, they have severely limited the group of polyphenolic compounds, omitting the most important ones. I believe that this part should be expanded to include other examples. The literature on this subject is very extensive and there is plenty to choose from.

Author Response

Thank you for your time reviewing and critiquing the manuscript. We have carefully addressed the comments and suggested revisions, as shown below:

Comment: While the first part is strong in my opinion, the rest of the manuscript is disappointing. The reader would expect more from the title. The authors limit themselves to only a few selected compounds, they have severely limited the group of polyphenolic compounds, omitting the most important ones. I believe that this part should be expanded to include other examples. The literature on this subject is very extensive and there is plenty to choose from.

Response: We thank the reviewer for this valuable comment. As suggested, we have added a section on four additional polyphenols (quercetin, ellagic acid, luteolin, and curcumin) (Pages 13-16). We’ve also added a section on celastrol (pages 19-20), and hydroxytyrosol (pages 20-21).